# GReNaDIne: A Data-Driven Python Library to Infer Gene Regulatory Networks from Gene Expression Data

**DOI:** 10.3390/genes14020269

**Published:** 2023-01-20

**Authors:** Pauline Schmitt, Baptiste Sorin, Timothée Frouté, Nicolas Parisot, Federica Calevro, Sergio Peignier

**Affiliations:** 1Univ Lyon, INSA-Lyon, INRAE, BF2i, UMR0203, F-69621 Villeurbanne, France; 2Univ Lyon, INRAE, INSA-Lyon, BF2i, UMR0203, F-69621 Villeurbanne, France

**Keywords:** bioinformatics, systems biology, gene regulatory network inference, gene expression, machine learning, ensemble learning, Python

## Abstract

*Context:* Inferring gene regulatory networks (GRN) from high-throughput gene expression data is a challenging task for which different strategies have been developed. Nevertheless, no ever-winning method exists, and each method has its advantages, intrinsic biases, and application domains. Thus, in order to analyze a dataset, users should be able to test different techniques and choose the most appropriate one. This step can be particularly difficult and time consuming, since most methods’ implementations are made available independently, possibly in different programming languages. The implementation of an open-source library containing different inference methods within a common framework is expected to be a valuable toolkit for the systems biology community. *Results*: In this work, we introduce GReNaDIne (Gene Regulatory Network Data-driven Inference), a Python package that implements 18 machine learning data-driven gene regulatory network inference methods. It also includes eight generalist preprocessing techniques, suitable for both RNA-seq and microarray dataset analysis, as well as four normalization techniques dedicated to RNA-seq. In addition, this package implements the possibility to combine the results of different inference tools to form robust and efficient ensembles. This package has been successfully assessed under the DREAM5 challenge benchmark dataset. The open-source GReNaDIne Python package is made freely available in a dedicated GitLab repository, as well as in the official third-party software repository PyPI Python Package Index. The latest documentation on the GReNaDIne library is also available at Read the Docs, an open-source software documentation hosting platform. *Contribution*: The GReNaDIne tool represents a technological contribution to the field of systems biology. This package can be used to infer gene regulatory networks from high-throughput gene expression data using different algorithms within the same framework. In order to analyze their datasets, users can apply a battery of preprocessing and postprocessing tools and choose the most adapted inference method from the GReNaDIne library and even combine the output of different methods to obtain more robust results. The results format provided by GReNaDIne is compatible with well-known complementary refinement tools such as PYSCENIC.

## 1. Introduction

Investigating the architecture of regulatory interactions between genes is a crucial step in gaining a deeper understanding of the complex and highly dynamic mechanisms that govern living organisms. The expression of numerous genes needs to be tightly regulated to control key biological processes, such as cell division and growth, tissue and organ differentiation, organisms’ development, and morphogenesis, as well as the response and adaptation to different environments [1,2,3,4]. The systems biology community often relies on gene regulatory networks (GRNs) to model these complex interactions between regulatory proteins, such as transcription factors (TFs) and their target genes (TGs). Recently, advancements in high-throughput sequencing techniques have made it possible to easily quantify gene expression from many experimental conditions and time-points. In parallel, these improvements have led to the development of computational methods that infer relationships between TFs and TGs to reconstruct GRNs. However, inferring GRNs from high-throughput gene expression data remains a challenging problem for the systems biology community, and a plethora of methods have been proposed to address this issue [5,6,7]. These inference techniques have been categorized in four major families [7]: (i) *data-driven methods* aim at analyzing gene expression data in order to assign a score to each possible link between a regulatory gene and its potential target gene and then selecting the most promising links; (ii) *probabilistic model-based methods* aim at representing a gene regulatory network as a probabilistic model (including Gaussian graphical models and Bayesian networks), and they rely on an experimental gene expression dataset to fit the parameters of the given probabilistic model; (iii) *dynamical model-based methods* aim at modeling a gene regulatory network as a dynamical system (including, for instance, dynamic Bayesian networks or systems of differential equations), and these methods are designed to analyze time-series gene expression data in order to estimate the dynamical model’s parameters; (iv) *multinetwork methods* aim at integrating different sources of information in the inference task, and they rely on a flexible gene regulatory network representation, allowing, for instance, network structural changes to cope with different biological conditions.

Among the different families of methods, GRN inference approaches based on the data-driven paradigm are among the most popular due to the fact of their ease of use, computational efficiency, and accuracy [7].

Early data-driven approaches aimed at inferring co-expression networks, assuming that the regulatory interactions between genes can be measured using correlation statistics [8] or information theory measures such as the mutual information (MI) [9]. Nevertheless, these methods only infer undirected links between pairs of related genes (i.e., the direction of the regulation between the genes is missing). Moreover, these techniques may include false positive links due to the presence of hidden confounding factors [10], and they may fail to capture complex regulatory patterns [11].

More recent data-driven techniques identify the regulators of a target gene as the subset of transcription factors, whose expressions contribute the most to the prediction of the expression level of the TG. These methods use a feature importance scoring procedure, training regression or classification algorithms to predict the expressions of TGs from those of their regulatory TFs [11,12,13]. Unlike co-expression methods, these methods rely on functional gene annotation to identify TFs, and they infer directed links between TFs and their TGs [11]. Given the relative scarcity of experimental conditions in a gene expression matrix, with respect to the high number of potential regulators, the GRN inference task is underdetermined and subject to the curse of dimensionality [14]. In this context, relying on prior information, such as gene annotation to reduce the number of potential regulators, aims at improving the quality of the inference task [15].

According to [16], each inference method has both intrinsic biases and advantages that make its application adapted to some particular datasets. Since no ever-winning inference method exists, users should apply different tools to analyze a dataset to choose an appropriate one. Given that most inference tools are deployed and distributed independently, this task could be particularly challenging and time consuming. Here we introduce GReNaDIne, a Python package implementing 18 data-driven GRN inference methods, 8 generalist preprocessing techniques for RNA-seq and expression microarray datasets, and 5 RNA-seq normalization techniques, as well as postprocessing techniques allowing for filtering of the most promising regulatory links, integration schemes for combining networks obtained by different inference tools in order to form robust ensembles of inference methods, and evaluation tools for comparing the inferred results with respect to gold standard benchmarks.

## 2. Implementation

GReNaDIne is an open-source software that consists of three separate modules (Figure 1) allowing to (i) preprocess gene expression data, (ii) score potential regulatory links with data-driven approaches, and (iii) select the most promising links to generate GRNs and evaluate the resulting GRNs. GReNaDIne is implemented as a Python 3 library and relies on widely used libraries, including Scikit-learn [17], NumPy [18], Pandas [19], and SciPy [20]. GReNaDIne is freely available in a dedicated GitLab repository (https://gitlab.com/bf2i/grenadine, accessed on 3 November 2022), as well as in Python Package Index (PyPi), the official third-party software repository for Python (https://pypi.org/project/GReNaDIne/, accessed on 3 November 2022). The latest documentation of the GReNaDIne library is also available in an open-source software documentation hosting platform called Read the Docs (https://grenadine.readthedocs.io/en/latest/, accessed on 3 November 2022).

### 2.1. Module 1: Preprocessing

As a preliminary step, through the first module, GReNaDIne aims at normalizing and standardizing the datasets. This first module integrates widely used RNA-seq normalization techniques to cope with library size biases (reads per million (RPM)), gene length biases (reads per kilobase (RPK)), or both problems simultaneously (reads per kilobase million (RPKM) and transcripts per kilobase million (TPM)). In addition, GReNaDIne includes a wrapper to use the DESeq2 [21] normalization function from Python. GReNaDIne also includes five discretization techniques for gene expression data: equal frequency discretization (EFD); equal width discretization (EWD); K-means discretization applied by rows (KMr) and columns (KMc); and the bidirectional K-means method also termed Bi-K-means (BKM) [22]. Finally, GReNaDIne incorporates three general standardization methods based on z-scores, namely, row-wise z-score (Zr), column-wise z-score (Zc), and polishing standardization (P) methods [23].

### 2.2. Module 2: Gene Regulatory Network Inference

Data-driven GRN inference methods score all possible regulatory links between TFs and TGs, based on their gene expression. Traditional GRN inference methods assume that the regulatory relationships between TFs and TGs can be inferred by measuring the correlation or the MI between their respective gene expression levels: GReNaDIne includes four co-expression methods based on the widely used Pearson correlation coefficient (PR), Spearman correlation coefficient (SR), Kendall tau statistic (KT), as well as a method termed context likelihood of relatedness (CLR) relying on a MI statistic. Moreover, GReNaDIne also includes 14 GRN inference methods based on well-known machine learning classification and regression algorithms. These methods aim at training a model (respectively, a classifier or a regressor) to predict the expression level of each TG from those of a set of TFs. Then, the importance of each TF to the prediction task is computed as a feature importance score. These scores are directly used as proxies to quantify the regulatory relation between each TF and its TGs. Regressors are directly trained on continuous gene expression data, while classifiers require the TG expression to be previously discretized. To do so, the discretization methods implemented in the first module can be used, and, in practice, the K-means discretization technique with five discrete levels of expression (i.e., five clusters) was applied successfully in [13]. GReNaDIne includes two methods based on support vector machines (SVMs) classifiers (c) and regressors (r), as described in [13]. It also incorporates eight methods based on ensembles of decision tree regressors and classifiers: AdaBoost (AB), gradient boosting (GB), random forest (RF), and eXtreme randomized trees (XRT) [11,13,24]. GReNaDIne also includes two methods based on regression stability selection criteria [12], called trustful inference of gene regulation using stability selection (TIGRESS) and stability randomized lasso (SRL). Moreover, GReNaDIne implements two novel methods based on Bayesian ridge regression (BRSr) [25] and complement naive Bayes classification (CNBc) [26]. In addition, module 2 of GReNaDIne allows to combine the results obtained from different methods in order to form an ensemble learning inference, as described in [27]. To do so, GReNaDIne includes different integration schemes. The general goal of these integration methods is to standardize the scores obtained from each inference method in order to make scores comparable between methods, and then to average the resulting scores to obtain a final ensemble score for each possible link between a TF and a TG. The differences between integration schemes are directly related to their underlying standardization technique. In practice, GReNaDIne includes six integration schemes based on the following standardizations: (i) *Z-score-full*: the distribution of scores obtained by a given method is standardized by means of a general z-score normalization, where each score is standardized by subtracting the mean of the distribution from all observed scores and then dividing by the standard deviation of the distribution; (ii) *Z-score-TG*: the distribution of scores obtained by a given method is standardized by means of a TG-centric z-score normalization, where each score is standardized by subtracting the mean of the distribution from scores of all regulatory links entering the corresponding TG, and then dividing by the standard deviation of this distribution; (iii) *Z-score-TF*: the distribution of scores obtained by a given method is standardized by means of a TF-centric z-score normalization, where each score is standardized by subtracting the mean of the distribution from scores of all regulatory links leaving the corresponding TF, and then dividing by the standard deviation of this distribution; (iv) *Rank-full*: each score is replaced by its corresponding rank in the distribution from all observed scores (1 representing the highest score); (v) *Rank-TG*: each score is replaced by its corresponding rank in the distribution of scores from all regulatory links entering the corresponding TG only; (vi) *Rank-TF*: each score is replaced by its corresponding rank in the distribution of scores from all regulatory links leaving the corresponding TF only.

Finally, the inference output format of GReNaDIne is compatible with the multinetwork PYSCENIC Python library [5]. Once a score matrix between TFs and their TGs is inferred, using any of the GReNaDIne methods, users can rely on the functions of the PYSCENIC library to refine GRN inferences that are solely based on gene expression data, by incorporating transcription factor binding site motifs analysis. The GReNaDIne GitLab repository includes a tutorial describing the integration with the PYSCENIC workflow on a *Mus musculus* single-cell RNA-seq dataset.

### 2.3. Module 3: Links Selection and Evaluation

After scoring all possible regulatory links between TFs and TGs, a classic procedure consists of selecting a subset of regulatory links with high scores to define putative GRNs. GReNaDIne includes functions that allow ranking of the possible regulatory links according to their scores, as well as functions that select the top-k links of the dataset and the top-k links involving a particular TF or TG. This module also includes some unsupervised tools that allow to compare the results obtained using different methods, based on principal component analysis and Jaccard index. Finally, the third GReNaDIne module includes some methods that allow to compute standard evaluation measures for binary classification (including the area under the precision recall curve (AUPR), the area under the receiver operating characteristic curve (AUROC, F1, Precision, and Recall scores), permitting to assess the predicted GRNs when gold standard datasets describing the validated regulatory links are available, such as in the DREAM5 benchmark. The methods implemented in this module are inspired by those described and used in [16]. In this evaluation framework, GRN inference is assessed as a binary classification task, where the goal is to classify pairs of TFs and TGs as true or false regulatory links. In this context, gold standard GRNs consist of experimentally verified regulatory links between TFs and TGs, considered as true interactions in the binary classification task. However, it is important to note that gold standard links are not a complete representation of all regulatory interactions in an organism, and missing links should not be considered as incorrect or false interactions, as they may represent true regulatory interactions that have not yet been experimentally verified. Therefore, any links involving a TF or a TG that has not been studied experimentally are excluded from the evaluation, and only links between an experimentally studied TF and TG, which are not present in the gold standards, are considered false interactions.

## 3. Results and Discussion

### 3.1. Evaluation Protocol

The DREAM5 evaluation framework [16] was used to assess the GReNADIne performances. This framework relies on three gold standard datasets from living organisms, namely, *Escherichia coli*, *Saccharomyces cerevisiae*, and *Staphylococcus aureus*, and a synthetic gold standard dataset. GRN inference has been evaluated as a binary classification task, which consists of predicting the presence of true regulatory links from gene expression, as described in Section 2. The task was assessed using the AUPR [28] and the AUROC [29] evaluation measures. The different evaluation datasets were made available online by their authors: https://www.synapse.org/#!Synapse:syn2820440/wiki/71024, accessed on 3 November 2022. The experiments presented in this article were executed on an Intel(R) Xeon(R) 2.40 GHz CPU, running Debian GNU/Linux 10, with a 120 Go RAM capacity.

### 3.2. Results

The GRN inference methods implemented in GReNaDIne provided results that were comparable or outperformed those obtained by the 35 DREAM5 competitors, as well as those obtained by the robust community approach that combined all participants’ results [16] (Figure 2). Interestingly, these inference methods tended to perform differently on each particular dataset, and there was no ever-winning technique. For instance, the inference methods based on ensembles of decision trees (i.e., ABr, ABc, GBr, GBc, RFr, RFc, XRTr, and XRTc) were comparable to the DREAM5 community for most datasets. The new approaches introduced in GReNaDIne based on SVMs (i.e., SVMr and SVMc), Bayesian ridge score (BRSr), and complement naive Bayes (CNBr) led to important gains for the real organisms’ datasets (i.e., *S. aureus*, *E. coli*, and *S. cerevisiae*) but exhibited a quality loss for the synthetic dataset. The methods based on correlation and MI led to comparable results for the real organisms’ datasets, while they exhibited poorer results for the synthetic dataset. These results are coherent with the conclusions presented in [16], which states that each inference method has both intrinsic advantages and biases that make its application suitable for some particular datasets.

For most of the inference methods, applying preprocessing techniques beforehand tended to have, on average, a rather small impact in terms of inference quality, considering both the AUROC and AUPR scores (Figure 3). However, as described in [16], the DREAM5 datasets were already normalized and filtered by means of robust multichip averaging, background adjustment, quantile normalization, probe set median polishing, and logarithmic scale normalization. This sophisticated preprocessing procedure is likely to reduce the impact of further preprocessing steps. The only method that exhibited a strong dependency with respect to the preprocessing step was the complement naive Bayes classifier (CNBc), which requires the equal frequency discretization (EFD) method to be applied beforehand (Figure 3). Regarding the other methods, the best preprocessing techniques were, on average, those ensuring that genes have comparable levels of expression, i.e., row z-score (Zr), equal frequency discretization (EFD), and row K-means (KMr) (Figure 3).

In addition, the ensembles of the GReNaDIne predictors described in [27], containing BRSr, an SVM-based method, and a method based on ensembles of trees or linear regressors (ensemble termed BRSr•SVM•Ens), and also including an extra correlation-based method (ensemble termed BRSr•SVM•Ens•Corr), revealed to be efficient and robust across different datasets, outperforming single methods as well as the robust community method presented in [12] (Figure 4). Moreover, the results obtained by the SVM•BRS•Ens ensembles, using different integration schemes, were compared in terms of the AUROC and AUPR scores on all DREAM5 datasets (Figure 5). The six integration schemes had suitable performances. However, it was observed that the rank-TF and Z-score-TF tended to show lower quality results when compared to Z-score-TG, rank-full, Z-score-full, and rank-TG. Therefore, it is recommended to use these latter integration schemes.

These results illustrate the functionalities made available in the GReNaDIne library and support the value of this package for the systems biology community interested in data-driven GRN inference.

### 3.3. Availability and Requirements

Project name: GReNaDIne—Gene Regulatory Network Data-driven Inference.

Project home page: https://gitlab.com/bf2i/grenadine, accessed on 3 November 2022.

PyPi repository web page: https://pypi.org/project/GReNaDIne/, accessed on 3 November 2022.

Project documentation: https://grenadine.readthedocs.io/en/latest/, accessed on 3 November 2022.

Programming language: Python 3.

License: GNU GPLv3.

## 4. Conclusions

The GReNaDIne toolbox represents a major methodological contribution to the field of systems biology. In addition to the classical co-expression network analysis, GReNaDIne also includes some more recent and complex GRN inference strategies, which rely on functional annotation of genes to identify transcription factors (TFs), infer directed links between TFs and their target genes (TGs), and predict the expression of TGs based on the expression of their TFs. The GReNaDIne package contains a battery of preprocessing tools that users can apply to treat their own experimental gene expression datasets or publicly available ones (e.g., NCBI Gene Expression Omnibus, https://www.ncbi.nlm.nih.gov/gds, accessed on 3 November 2022). In addition, users can rely on ad hoc gene annotations or publicly available ones (e.g., AnimalTFDB4 Animal Transcription Factor Database, http://bioinfo.life.hust.edu.cn/AnimalTFDB4, accessed on 3 November 2022) in order to run any of the 18 GReNaDIne methods and perform gene regulatory networks inference in their preferred biological system. Users can thus rely on this toolbox to choose the most adapted method to their dataset, within the same framework, and even combining the output of different methods to obtain robust and accurate results.

## Figures and Tables

**Figure 1 genes-14-00269-f001:**
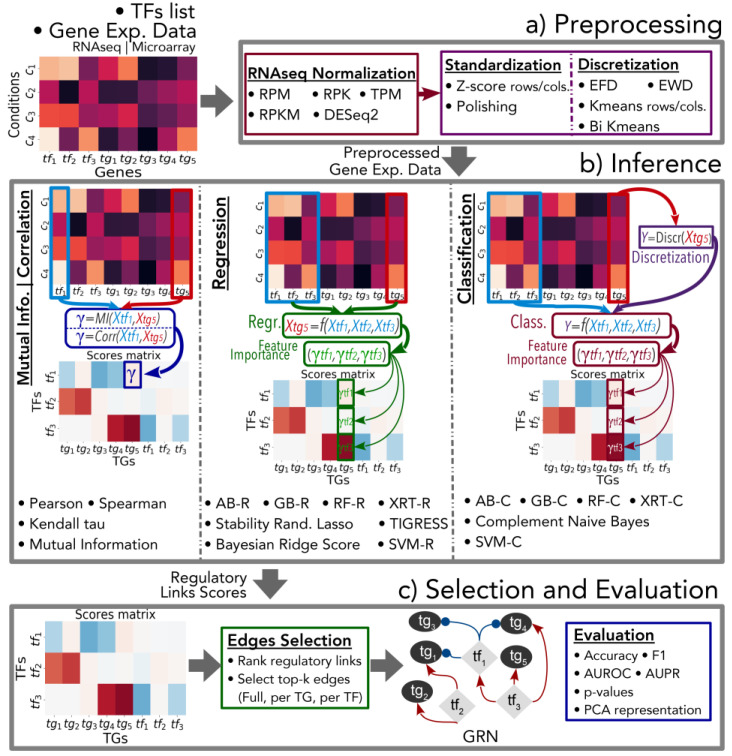
The GReNaDIne GRN Inference workflow is organized in three modules: (**a**) Gene expression preprocessing, including RNA-seq normalization, standardization, and discretization techniques. (**b**) GRN data-driven inference scoring methods, including techniques based on MI and correlation scores, methods based on regression algorithms, and techniques based on classification algorithms. This second module also incorporates some integration schemes to combine results from different methods to form ensembles. (**c**) Postprocessing regulatory edges selection tools and GRN evaluation methods. The GRN inference workflow of GreNaDIne simply requires as inputs a gene expression matrix and facultatively a list of regulatory genes (e.g., TFs).

**Figure 2 genes-14-00269-f002:**
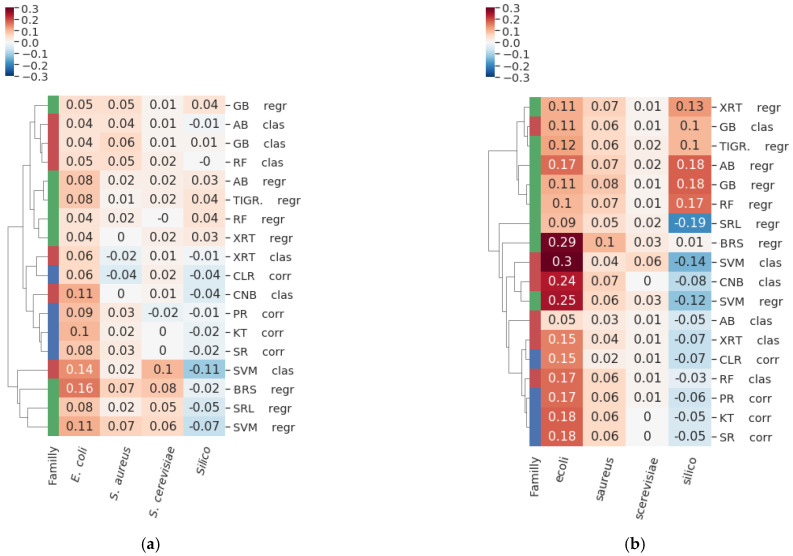
Cluster maps representing the gain in (**a**) AUROC and (**b**) AUPR values for each inference methods (rows) without using any preprocessing techniques on each benchmark dataset (columns), with respect to the AUROC or AUPR reference score of the DREAM 5 community approach. The family of each method is reported in colors in the left column: regression in green, classification in red, and correlation/MI in blue. The GReNaDIne inference methods exhibited comparable and even better results than those obtained by the DREAM5 community approach. The inner biases and advantages of each method make it suitable for some particular datasets; indeed, the methods performed differently on each particular dataset, and no ever-winning method existed.

**Figure 3 genes-14-00269-f003:**
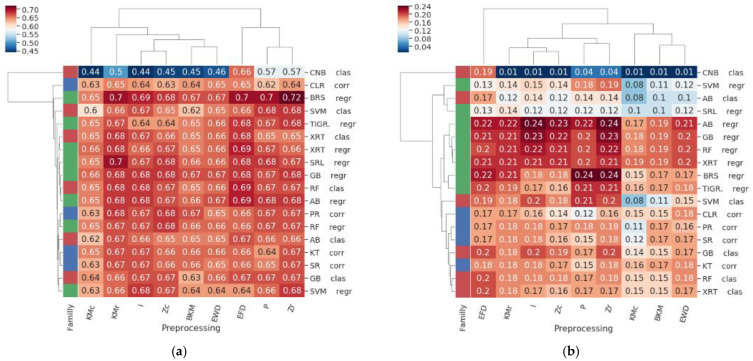
Cluster maps representing the (**a**) AUROC and (**b**) AUPR values for each combination of inference methods (rows) and preprocessing technique (columns); notice that column I represents the identity (i.e., no preprocessing technique applied). The family of each method is reported in colors in the left column: regression in green, classification in red, and correlation/MI in blue. Preprocessing techniques that ensure that genes exhibit comparable levels of expression (i.e., row z-score, EFD, and row K-means) led to better performances on average.

**Figure 4 genes-14-00269-f004:**
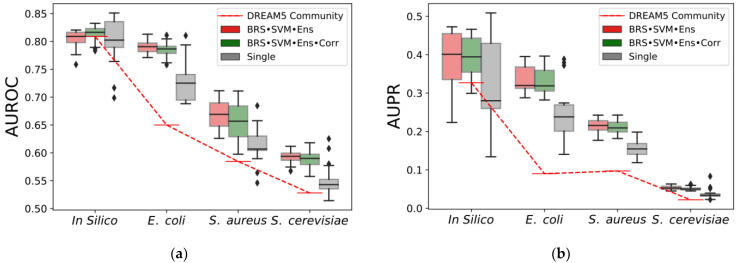
(**a**) AUROC and (**b**) AUPR scores obtained by ensembles of the inference methods (i.e., BRS•SVM•Ens and BRS•SVM•Ens•Corr), single methods, and DREAM5 community. The ensembles of GReNaDIne that contained BRSr, an SVM-based method, as well as a method based on ensembles of trees or linear regressors (ensemble termed BRSr•SVM•Ens), and also including an extra correlation- or MI-based method (ensemble termed BRSr•SVM•Ens•Corr), revealed to be efficient and robust across different datasets, outperforming single methods as well as the robust DREAM5 community method, with respect to both the AUROC and AUPR scores.

**Figure 5 genes-14-00269-f005:**
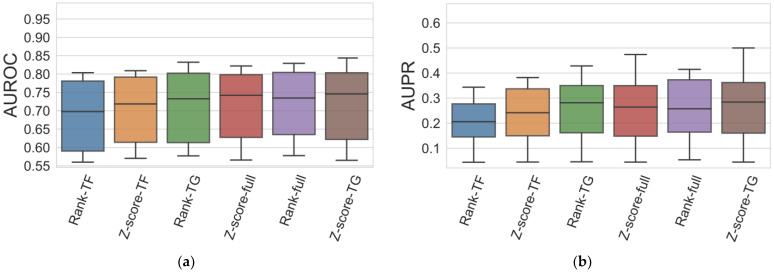
Boxplots representing the average (**a**) AUROC and (**b**) AUPR obtained by the SVM•BRS•Ens ensembles, with different integration schemes, on all DREAM5 datasets. The integration schemes are arranged in ascending order based on their average AUROC and AUPR scores. All integration schemes had suitable results, but rank-TF and Z-score-TF tended to exhibited lower results compared to Z-score-TG, rank-full, Z-score-full, and Rank-TG. Therefore, it is suggested to use these latter integration schemes.

## Data Availability

The DREAM5 datasets analyzed in the current study were made available online by their authors at https://www.synapse.org/#!Synapse:syn2820440/wiki/71024, accessed on 3 November 2022. The open-source GReNaDIne Python package is made freely available in a dedicated GitLab repository (https://gitlab.com/bf2i/grenadine, accessed on 3 November 2022), as well as in the official third-party software repository PyPI Python Package Index (https://pypi.org/project/GReNaDIne/, accessed on 3 November 2022). The latest documentation of the GReNaDIne library is also available in the open-source software documentation hosting platform Read the Docs (https://grenadine.readthedocs.io/en/latest/, accessed on 3 November 2022).

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
