# Peer review of "GReNaDIne: A Data-Driven Python Library to Infer Gene Regulatory Networks from Gene Expression Data"

_genes, 2023, doi:10.3390/genes14020269_

Round 1
Reviewer 1 Report
Schmitt and colleagues introduced a Python package named GReNaDine for integration analysis of inferring gene regulatory networks from high-throughput gene expression data (such as RNA-seq). The package is helpful to the bioinformatics and gene expression research community, and the code is open-source hosted and well-written. Python-based is an excellent choice. The documentation for the package is well organized, and example data for the test are well provided. The manuscript is presented in a straightforward and clear way.
Reviewer 2 Report
Generally speaking every authentic package can potentially find their users. In this case I would suggest to be more clear regarding the applicability of the package. Since the input data are RNAseq or Expression microarrays orients might think of the differential gene expression analysis, which is not the case here. To the contrary, the package is dedicated to the to such a mainstream application as a coexpression networks. In this regard I would avoid references to the transcription factors and targets, because this package (to my understanding) does not use such information from available databases. If i am wrong, then I would like to ask authors to explain more thoroughly how information about transcription factors and targets is used in their application.
Round 2
Reviewer 2 Report
Authors responded on my concerns. I have no further comments.